# Effects of Grape Polyphenols on the Life Span and Neuroinflammatory Alterations Related to Neurodegenerative Parkinson Disease-Like Disturbances in Mice

**DOI:** 10.3390/molecules25225339

**Published:** 2020-11-16

**Authors:** Maria A. Tikhonova, Nadezhda G. Tikhonova, Michael V. Tenditnik, Marina V. Ovsyukova, Anna A. Akopyan, Nina I. Dubrovina, Tamara G. Amstislavskaya, Elena K. Khlestkina

**Affiliations:** 1Federal Research Center “Institute of Cytology and Genetics”, Siberian Branch of the Russian Academy of Sciences, 630090 Novosibirsk, Russia; tikhonovama@physiol.ru (M.A.T.); khlest@bionet.nsc.ru (E.K.K.); 2Federal State Budgetary Scientific Institution “Scientific Research Institute of Neurosciences and Medicine” (SRINM), 630117 Novosibirsk, Russia; m.v.tenditnik@physiol.ru (M.V.T.); maryov@ngs.ru (M.V.O.); annaaleksanovna@mail.ru (A.A.A.); dubrov@physiol.ru (N.I.D.); amstislavskaya@yandex.ru (T.G.A.); 3N.I. Vavilov All-Russian Institute of Plant Genetic Resources, 190000 St. Petersburg, Russia

**Keywords:** flavonoids, cognition, passive avoidance test, memory extinction, mice, microglia, neuroprotection

## Abstract

Functional nutrition is a valuable supplementation to dietary therapy. Functional foods are enriched with biologically active substances. Plant polyphenols attract particular attention due to multiple beneficial properties attributed to their high antioxidant and other biological activities. We assessed the effect of grape polyphenols on the life span of C57BL/6 mice and on behavioral and neuroinflammatory alterations in a transgenic mouse model of Parkinson disease (PD) with overexpression of the A53T-mutant human α-synuclein. C57BL/6 mice were given a dietary supplement containing grape polyphenol concentrate (GPC—1.5 mL/kg/day) with drinking water from the age of 6–8 weeks for life. Transgenic PD mice received GPC beginning at the age of 10 weeks for four months. GPC significantly influenced the cumulative proportion of surviving and substantially augmented the average life span in mice. In the transgenic PD model, the grape polyphenol (GP) diet enhanced memory reconsolidation and diminished memory extinction in a passive avoidance test. Behavioral effects of GP treatment were accompanied by a decrease in α-synuclein accumulation in the frontal cortex and a reduction in the expression of neuroinflammatory markers (IBA1 and CD54) in the frontal cortex and hippocampus. Thus, a GP-rich diet is recommended as promising functional nutrition for aging people and patients with neurodegenerative disorders.

## 1. Introduction

There is growing evidence that diets rich in polyphenols decrease the risk of chronic diseases such as obesity, diabetes, heart disease, and cancer [1,2,3,4,5]. Polyphenols are the most abundant group of biologically active molecules and natural antioxidants that can protect human cells from oxidative damage thereby reducing the risk of developing various degenerative diseases associated with oxidative stress. The main sources of polyphenols are fruits, tea, coffee, cacao, and grapes. These compounds are also found in vegetables and seeds of different crops, but in lower concentrations [6,7]. In plants, polyphenols are secondary metabolites participating in plant protection from a wide range of biotic and abiotic stress factors [8,9].

Even with a standard diet, we consume about 1 g of polyphenols daily, which is 10 and 100 times higher than the daily doses of vitamins C and E (as well as carotenoids), respectively [10]. In addition to antioxidant activity, polyphenols depending on their chemical structure, manifest a wide range of biological activities, such as anti-inflammatory, capillary-strengthening, hepatoprotective, diuretic, antiallergic, antimicrobial and antitumor properties [5,11,12,13].

There is a special interest in reusing winemaking waste, such as grape seed, fruit skin or vine. These waste products are rich in polyphenols, specifically, catechins, quercetin, anthocyanins, proantocyanidins, phenolic acids, and resveratrol [14,15] having most of the activities described above [5,12,13], so they may be used to produce valuable extracts. Furthermore, the photoprotective effect of polyphenolic compounds extracted from red grapes was observed during in vitro human cell culture studies [7]. Red grapes effectively inhibit UV-A-induced synthesis of type III collagen both at the RNA and protein levels. This confirms the potential of grape polyphenols in slowing down the skin photoaging [7]. In addition, polyphenols can overcome the blood–brain barrier and produce a neuroprotective effect on the central nervous system [16,17]. Hence, the application of grape-derived polyphenols as an adjunctive treatment paradigm to prevent neuropathologies including such neurodegenerative disorders as Alzheimer and Parkinson diseases is widely discussed [18]. Parkinson disease (PD) is the second most common neurodegenerative disorder, placing huge economic and social burdens on societies all over the world. The main risk factor for the development of PD is aging. Current therapies for PD are symptomatic and limited they do not cope with the disease onset or progression as they do not address multiple overlapping mechanisms involved in the PD pathogenesis. Moreover, PD patients develop drug tolerance and suffer from serious side effects of the drugs due to uninterrupted long-term treatment. Nutritional supplementation with polyphenols is regarded as a promising prophylactic treatment for neurodegenerative disorders, as it potently and simultaneously targets inflammatory and oxidative pathways [18]. It is noteworthy that a recent study by Ben Youssef et al. [19] evidenced the neuroprotective effect of grape seed and skin extract on a mouse PD model induced by 6-OHDA neurotoxin through reducing apoptosis, oxidative stress, and inflammation. Pathological aggregation and accumulation of α-synuclein in neurons and Lewy bodies appear to play a core role in the pathogenesis of PD [20]. However, the potential impact of grape polyphenols on this mechanism is scantily studied.

In the current study, we assessed the effect of grape polyphenols on the life span of C57BL/6 mice and on behavioral and neuroinflammatory alterations in a transgenic mouse model of PD with overexpression of the A53T-mutant human α-synuclein.

## 2. Results

### 2.1. Diet Tolerance and the Effects on Life Expectancy and Body Weight Gain

In mice of the C57Bl/6 strain born and reared at the conventional animal facility, grape polyphenol concentrate (*GPC*) significantly influenced the cumulative proportion of surviving (*p* < 0.01; Figure 1A) and substantially augmented the average life span (*p* < 0.01; Figure 1B).

The last mouse in the experiment was a *GPC*-treated male who died at the age of 1031 days. Moreover, in two-year-old mice the general condition of *GPC*-treated animals appeared to be better than of the controls, including the state of their fur and eyes (Figure 1C). Mice of the B6.Cg-Tg (Prnp-SNCA*A53T)23Mkle/J strain (further mut(PD)), a genetic model of PD, and control wild-type (WT) mice that had been born and reared under SPF conditions also endured the *GPC*-supplemented diet well. No significant difference was found between the groups in body weight gain after four months of the experiment (Figure 2).

### 2.2. Behavioral Effects

#### 2.2.1. The Open Field Test

An open field test was performed to assess general locomotion, vertical locomotor and exploratory activity, anxiety, and emotionality in mice (Table 1).

When testing a PD model, two-way ANOVA followed by an LSD post-hoc test revealed a significant effect of the genotype on the distance travelled. Mut(PD) mice had higher horizontal locomotion than WT mice, which is quite in line with previous studies on this PD model [21,22,23]. The other measured parameters were not significantly affected by the genotype factor or the diet factor or their interaction.

#### 2.2.2. The Passive Avoidance Test

We recorded a significant effect of the repeated measures (learning) factor (F(1, 28) = 96.4, *p* < 0.001) on the step-through latency when evaluating contextual memory retrieval in mice (Figure 3A).

Latency to enter a dark compartment during training (before the foot shock) did not differ significantly among the experimental groups. As evidence of learning and acquisition of the conditioned passive avoidance reaction on testing day, 24 h after receiving the foot shock, mice of all groups demonstrated increased step-through latencies, often ~10-times greater than latencies on the training day. However, memory extinction was influenced not only by the repeated measures (time) factor (F (11, 297) = 14.6, *p* < 0.001)) but also by the interactions between the genotype and diet factors (F (1, 27) = 6.03, *p* < 0.05) and between the repeated measures and diet factors (F (11, 297) = 3.9, *p* < 0.001) (Figure 3B). With exposure to the context in the absence of additional shocks, the fear response gradually diminishes which is called memory extinction [24]. In the WT control, WT + GPC, and mut(PD) control groups, the values of step-through latency remained significantly increased for seven, seven, and two days, respectively, compared to the training day. A significant decrease in step-through latency was observed to start from the 6th, 8th, and 4th day of the extinction phase compared to the test day in WT control, WT + *GPC*, and mut(PD) control group, respectively. Hence, extinction was more pronounced in the mut(PD) control group. At the same time, the values of step-through latency stayed significantly increased for ten days of the extinction phase as compared to the training day in a mut(PD) + *GPC* group. We failed to observe a substantial reduction in step-through latency in mice of the mut(PD) + *GPC* group within ten days of the extinction phase. Thus, the mut (PD) mice demonstrated *GPC* enhanced memory reconsolidation and diminished memory extinction.

### 2.3. Immunohistochemical Analysis

First, we evaluated the accumulation of human α-synuclein in the mouse brain. We detected immunofluorescence against human α-synuclein only in the frontal cortex of seven-month-old transgenic mut(PD) mice (Figure 4B). Both the genotype (F (1, 8) = 35.2, *p* < 0.001) and diet (F (1, 8) = 7.5, *p* < 0.05) factors had a significant effect on α-synuclein accumulation in the 2nd layer of the frontal cortex (Figure 4A).

Dietary supplementation with *GPC* significantly reduced the accumulation of human α-synuclein in the frontal cortex of mut(PD) mice (*p* < 0.01). Indices of neuroinflammation (microglial marker IBA1, Figure 4C; inflammatory marker CD54, Figure 4D) were also increased in the frontal cortex of transgenic mut(PD) mice, while *GPC* treatment significantly suppressed them back to the levels of WT mice. Similar effects on the expression of the inflammatory markers were recorded in the hippocampus (Figure 5; Appendix A: Appendix A).

The diet factor (F (1, 8) = 23.5, *p* < 0.01) produced a significant effect on the expression of IBA1 in the CA1 hippocampal area. The parameter was markedly decreased in both WT (*p* < 0.05) and mut(PD) (*p* < 0.01) mice by *GPC* supplementation (Figure 5A). The expression of CD54 was significantly influenced by the diet factor (F(1, 8) = 5.6, *p* < 0.05) in the CA3 hippocampal area (Figure 5E) and by the genotype factor (F(1, 8) = 11.96, *p* < 0.01), diet factor (F(1, 8) = 13.1, *p* < 0.01), and interaction between the factors (F(1, 8) = 6.8, *p* < 0.05) in the dentate gyrus of the hippocampus (Figure 5F). Dietary supplementation with *GPC* significantly reduced the CD54 expression in the CA3 area (*p* < 0.05) and dentate gyrus (*p* < 0.01) in mut(PD) mice.

## 3. Discussion

### 3.1. Safety of Long-Term Supplementation with Grape Polyphenols and Their Effect on Life Span in Mice

Pallauf et al. [25] reviewed the data on life span extension by flavonoids in worms, flies and mice, describing possible mechanisms that may underpin longer life spans of model organisms treated with flavonoids: energy-restriction-like effects, inhibition of insulin-like-growth-factor signaling, induction of antioxidant activity, hormesis and antimicrobial properties.

In the experiments, with mice, their life span increased when the diets were supplemented with combined extracts of blueberry, green tea and pomegranate powder [26] or green tea polyphenols containing approximately 70% epigallocatechins, epicatechins and gallocatechins [27]. In other studies, however, experimenting with blackcurrant juice (containing anthocyanins, quercetin and quercetin glycosides), epimedium flavonoids (containing 20% icariin) [28], green tea extracts [29,30], blueberry extract [30], triple combination of green tea extract, black tea extract and morin [30] or pomegranate powder [30] failed to induce a longer life span in mice. Overall, several issues should be taken into account when the potential effect of flavonoids on animal or human life span in discussed. When different flavonoids are consumed simultaneously, they may have an additive or even synergistic impact on life expectancy. Moreover, other compounds in extracts may either attenuate or improve the life-extending effect of flavonoids. Besides, synergy was observed between polyphenols, drugs, and hormones [16]. The duration of treatment and the age when the supplementation starts also matter and may lead to different outcomes.

In our study, grape polyphenol concentrate (containing a wide range of anthocyanins, flavan-3-ols, hydroxycinnamic acids, high molecular weight, oligomeric and condensed procyanidins plus 121.2 mg/L, of quercetin, 46.0 mg/L of quercetin-3-O-glycoside,928.4 mg/L of gallic acid and 5.6 mg/L of trans-resveratrol; Appendix A: Appendix A) increased life span (Figure 1). A conclusion can be made that *GPC* was safe for mice during almost life-long supplementation starting from an early age. The long-term treatment did not produce any visible harmful effects on the general condition of the animals, nor disturbed their behavioral performance or cognitive function. This observation is quite important, since a preventive long-term intervention at the asymptomatic preclinical and early stages of the disease progression is considered the most promising in the management of neurodegenerative disorders [31].

### 3.2. Polyphenols Attenuate Neuropathological Changes Associated with Aging and PD-Like Disturbances

Our experiments with a transgenic PD model, demonstrated the grape polyphenols enhanced memory reconsolidation and abated memory extinction in the passive avoidance test (Figure 3). It should be mentioned that the open field test revealed no significant alterations in the *GPC*-treated groups. Hence, the observed effect of *GPC* on cognitive functions was specific and did not depend on general changes in locomotor or exploratory behavior. The beneficial effect of grape polyphenols on cognitive functions agrees well with the previous findings on the restoration of impaired cognition in the mouse models of aging or Alzheimer disease [32,33,34,35].

Low bioavailability of flavonoids questioned their direct effects on the central nervous system. However, when their bioactive metabolites are taken into account, including those associated with the activity of the gut microbiota and interaction products, bioavailability appears to be much higher [36]. For example, anthocyanins and their metabolites were found in almost all organs and tissues including the brain of animals fed with anthocyanin-rich feeds. The latter observation attests to active absorption and the ability to overcome the blood–brain barrier [37,38,39]. Hence, the results obtained on biological activity and impact mechanisms of the intact compounds produced in vitro should be interpreted with a certain caution and need to be confirmed by in vivo findings. Moreover, the dosage is also an essential issue. For example, high doses of resveratrol applied to overcome its low bioavailability caused various side effects [40]. Moreover, some components of the *GPC* mixture, such as resveratrol and epigallocatechin-3-gallate, are regarded as pan-assay-interference (PAINS) compounds [41]. Hence, their effect on an organism might be nonspecific, including cell membrane perturbations, rather than specific protein binding [42], especially when high supraphysiological doses are applied.

Nevertheless, preclinical animal studies revealed that grape polyphenols might affect certain pathogenetic mechanisms involved in the aging-related cognitive decline and neurodegenerative disorders, such as neuroinflammatory response, oxidative stress, protein homeostasis, and apoptotic signaling [17,32,33]. A systematic review of 43 publications performed by de Andrade Teles et al. [43] summarized that the main targets of action for the flavonoid-based PD therapy were the reduction of the cellular oxidative potential and activation of neuronal death mechanisms. Strathearn et al. [44] suggested that anthocyanin- and proanthocyanidin-rich plant extracts could alleviate PD-induced neurodegeneration by enhancing the mitochondrial function.

Although pronounced motor disturbances occur in transgenic mice with overexpression of mutant human α-synuclein at the age of 9–13 months [23], certain behavioral and cognitive alterations appear at early stages of the pathology course including memory deficit [22]. Those non-motor symptoms are associated with the accumulation of α-synuclein and neuroinflammation in the forebrain regions. Indeed, we revealed the deposits of α-synuclein and enhanced expression of inflammatory markers (IBA1 and CD54) in the frontal cortex of transgenic mice. *GPC* supplementation significantly decreased the α-synuclein accumulation and reduced the expression of the neuroinflammatory markers in the frontal cortex (Figure 4). We did not find any d α-synuclein deposition in the hippocampus while the neuroinflammatory response was less pronounced. However, *GPC* treatment produced e similar effects attenuating the inflammatory markers in the hippocampal regions (Figure 5). Many studies reported on the anti-inflammatory effect of polyphenols, including the recent study of a mouse PD model induced by 6-OHDA neurotoxin [20]. At the same time, the effect of polyphenols on α-synuclein aggregation and neurotoxicity was observed in vitro in cellular models [45,46]. Specifically, the major metabolite among anthocyanins cyanidin 3-glucoside inhibited aggregation and fibril formation of α-synuclein [47,48]. The present study confirmed the neuroprotective activity of grape polyphenols against α-synuclein in vivo. Thus, *GPC* supplementation has a potential therapeutic effect in preventing and treating Parkinson disease.

## 4. Materials and Methods

### 4.1. Experimental Animal and Design

Experiments were performed using male mice: (1) non-SPF mice of the C57Bl/6 strain born and reared in a conventional animal facility at the Federal State Budgetary Scientific Institution “Scientific Research Institute of Neurosciences and Medicine” (SRINM; former Scientific Research Institute of Physiology and Basic Medicine) (Novosibirsk, Russia); (2) mice of the B6.Cg-Tg(Prnp-SNCA*A53T)23Mkle/J) strain (hereinafter: mut(PD)) and control WT mice acquired from the SPF-vivarium of the Institute of Cytology and Genetics SB RAS (Novosibirsk, Russia). Hemizygous mut (PD) mice were produced by inserting the human A53T missense mutant form of alpha-synuclein cDNA in the mouse genome downstream of a mouse prion Prnp promoter [49].

Animals were housed in groups of 4–5 animals per cage (40 × 25 × 15 cm) under standard conditions (light–dark cycle: 14 h light and 10 h dark (lights off at 15:00); temperature: 18–22 °C; relative humidity: 50–60%). All experimental procedures were carried out in accordance with the guidelines of the NIH Guide for the Care and Use of Laboratory Animals and were approved by the Institutional Animal Care and Use Committee of the SRINM. Every effort was made to minimize the number of animals used and their suffering.

In each experiment, mice of each strain were subdivided into two groups and prescribed one of the following diets. The mice of the control groups received a standard granulated chow for laboratory mice (Ssniff R/M-H V1534-300, Soest, Germany) and pure water (Rosinka, Novosibirsk, Russia) ad libitum. The mice of the *GPC* groups were given the standard chow and the grape polyphenol aqueous solution of the concentrate *Enoant* (produced by RESSFOOD, Company, Yalta, Russia) [50] ad libitum. *Enoant* was diluted with the pure water to a concentration of 0.5–0.8% taking into account a mouse body weight and liquid consumption to provide the daily averaged dosage of 1.5 mL/kg. Fresh solution was prepared every other day. To adjust the dosage, liquid consumption per cage was measured daily and mice were weighed weekly. Concentrations of polyphenolic compounds in *Enoant* are described in Appendix A: Appendix A.

The experiment with non-SPF mice of the C57Bl/6 strain born and reared at the conventional animal facility started when mice reached an age of 6–8 weeks and lasted until the death of the last mouse (control, *n* = 15; *GPC*, *n* = 8). Mut(PD) mice, a genetic model of PD, and control WT mice born and reared under SPF conditions were fed with the *GPC* supplement since the age of six weeks (WT control, *n* = 9; WT + GPC, *n* = 9; mut(PD) control, *n* = 8; mut(PD) + *GPC*, *n* = 9). After four months of *GPC* feeding mut(PD) and WT mice were tested for behavior and then sacrificed for further immunohistochemical analysis of their brains.

### 4.2. Behavioral Tests

Each animal was handled for 5 min/day on three consecutive days, prior to being taken into the experiment. Open field and passive avoidance tests were performed. Observations were made during the dark phase between 15:00 and 22:00 h. For behavioral testing, the animals were placed individually in a clean cage (25 × 40 × 20 cm), and transported to a dim observation room (28 lux of the red light) with sound isolation reinforced by a masking white noise of 70 dB. Performance in the behavioral tests was monitored using a video camera Panasonic WV-CL930 (Panasonic System Networks Suzhou Co.,Ltd., Suzhou, China) positioned above an apparatus and processed with original EthoVision XT software (Noldus, Wageningen, The Netherlands). The test equipment was cleaned using 20% ethanol and thoroughly dried before each test trial.

#### 4.2.1. The Open Field Test

This test was carried out in an apparatus with a square arena (40 × 40 cm) and plastic walls 37.5 cm high brightly lit from above (1000 lux). A mouse was placed in the center of the arena, and its movements were recorded for 10 min. The following parameters were assessed: general locomotion (distance travelled in cm); vertical locomotor and exploratory activity (number of rearing); anxiety (time spent in the central part of the arena); and emotionality (number of defecations).

#### 4.2.2. The Passive Avoidance Test

Training on the passive avoidance reaction was performed by a standard single-session method in an experimental chamber with dark and light compartments and an automated Gemini Avoidance System apparatus (San Diego Instruments, San Diego, CA, USA)) as described in detail earlier [51]. The Gemini software automatically recorded the latency of the transfer to the dark compartment and the data of testing served as a measure of acquisition of the conditioned passive avoidance reaction. Memory extinction was measured during the next ten days.

### 4.3. Immunohistochemical Analysis

On the day of euthanasia, mice were culled with CO_2_. The animals were perfused transcardially with phosphate-buffered saline (PBS) followed by 4% paraformaldehyde in PBS, then the brains were rapidly excised and postfixed in PBS containing 30% sucrose at 4 °C until further neuromorphological analysis. The analysis was performed on 30-μm-thick cryosections according to a protocol described in detail previously [18]. Coronal slices along the frontal cortex (AP: 2.93 to −2.45 mm) or hippocampus (AP: −2.03 to −2.15 mm) of each mouse brain were made. We applied a rabbit polyclonal antibody (NB110-61645, 1:1000 dilution, Novus Biologicals, Littleton, CO, USA) as the primary antibody to detect human α-synuclein, a goat polyclonal antibody (NB100-1028, 1:200 dilution, Novus Biologicals, Littleton, CO, USA) as the primary antibody to detect the AIF-1/IBA1 microglial marker, or a rat monoclonal antibody (catalog # 16-0542-81, 1:300 dilution, Invitrogen, Carlsbad, CA, USA) as the primary antibody to detect the CD54(ICAM-1) inflammatory marker. A fluorescently labeled (Alexa Fluor 488-conjugated) goat anti-rabbit IgG antibody (ab150077, 1:600 dilution, Abcam, Cambridge, UK), Alexa Fluor 488-conjugated donkey anti-goat IgG antibody (ab150129, 1:200 dilution, Abcam, Cambridge, UK), or Alexa Fluor 594-conjugated goat anti-rat IgG antibody (ab150160, 1:500 dilution, Abcam, Cambridge, UK) served as the secondary antibodies, respectively. Fluorescent images were finally obtained by means of an Axioplan 2 (Carl Zeiss) imaging microscope and then analyzed in Image Pro Plus Software 6.0 (Media Cybernetics, Inc., Rockville, MD, USA). Fluorescence intensity was measured as background-corrected optical density (OD) with subtraction of staining signals of the non-immunoreactive regions in the images converted to grayscale. The area of interest was 7423 μm^2^ (IBA1 or CD54) or 30,014 μm^2^ (α-synuclein) in the frontal cortex; 19,353 μm^2^, 26,100 μm^2^, and 50,868 μm^2^ in the hippocampal CA1, CA3 areas, and dentate gyrus, respectively.

### 4.4. Data Analysis

Survival analysis was performed using Gehan’s Wilcoxon test and presented as a Kaplan–Meier diagram; average life span for each diet was compared with the Mann–Whitney U-test. The results on the PD model were presented as mean ± SEM and compared using a two-way ANOVA followed by Fisher LSD post-hoc test. The independent variables for the two-way ANOVA were Genotype (WT or mut(PD)) and Diet (control or *GPC*). Repeated Measures ANOVA followed by Fisher LSD post-hoc comparison was applied to analyze the data of the passive avoidance test with Genotype and Diet as between-subject variables and Time (Training, Test, or Extinction days) as a repeated measure. The level of significance was defined as *p* < 0.05. STATISTICA 10.0 software was used to perform all statistical analyses.

## 5. Conclusions

*GPC* proved safe for mice during almost life-long supplementation starting from an early age. The long-term *GPC* feeding did not produce any visible harmful effects on the general condition of the animals, nor disturbed their behavioral performance or cognitive function. Hence, such treatment seems to be applicable as a preventive long-term intervention at the asymptomatic preclinical and early stages of neurodegenerative diseases. *GPC* affected the cognitive function in transgenic PD mice by modulating their memory-specifically, by enhancing memory reconsolidation and abating memory extinction. The behavioral and cognitive effects of *GPC* were accompanied by a reduction in α-synuclein accumulation in the frontal cortex and neuroinflammatory response in the frontal cortex and hippocampus. Our findings confirmed the neuroprotective activity of grape polyphenols against α-synuclein in vivo. Thus, a diet supplemented with grape polyphenol concentrate is suggested as promising functional nutrition for aging people and patients with neurodegenerative disorders, especially Parkinson’s disease.

## Figures and Tables

**Figure 1 molecules-25-05339-f001:**
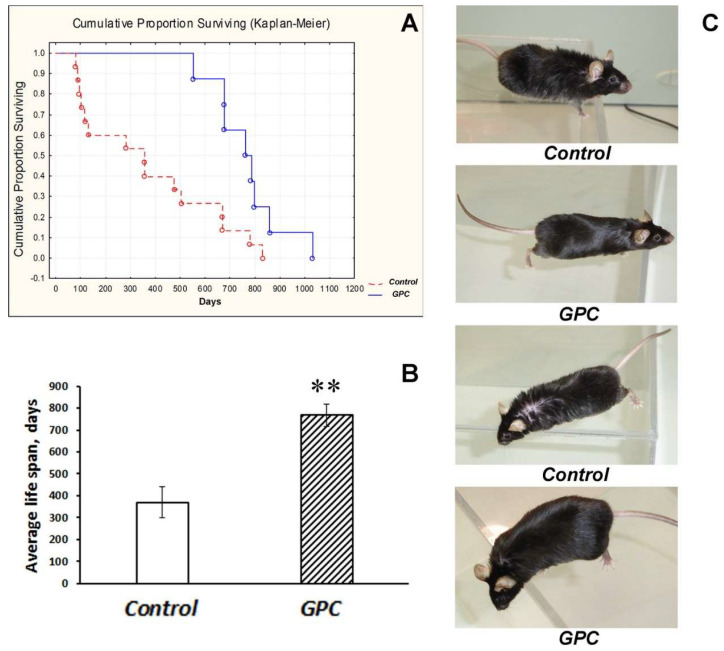
The effect of dietary supplementation with *GPC* on the cumulative proportion of surviving (**A**), average life span (**B**), and general condition (**C**) in mice of the C57Bl/6 strain. The data are presented as a Kaplan–Meier diagram (**A**) or the means ± SEMs (**B**) of the values obtained in an independent group of animals (*n* = 8–15 per group). Statistically significant differences: ** *p* < 0.01 vs. controls. (**C**) photographs of control and *GPC*-treated mice at the age of two years.

**Figure 2 molecules-25-05339-f002:**
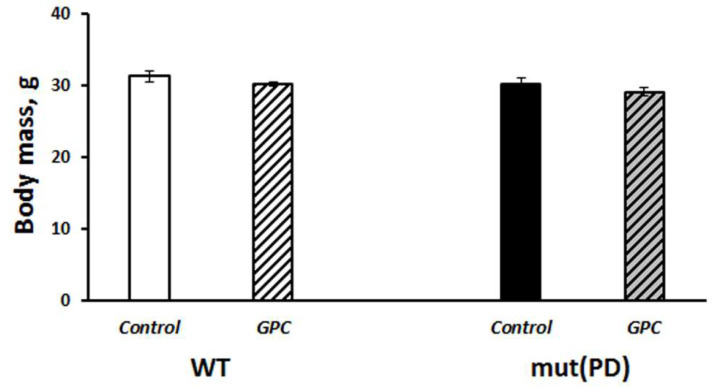
Effects of the overexpression of A53T-mutant α-synuclein and dietary supplementation with *GPC* for four months on body weight gain in mice.The data are expressed as the means ± SEMs of the values obtained in an independent group of animals (*n* = 5–9 per group).

**Figure 3 molecules-25-05339-f003:**
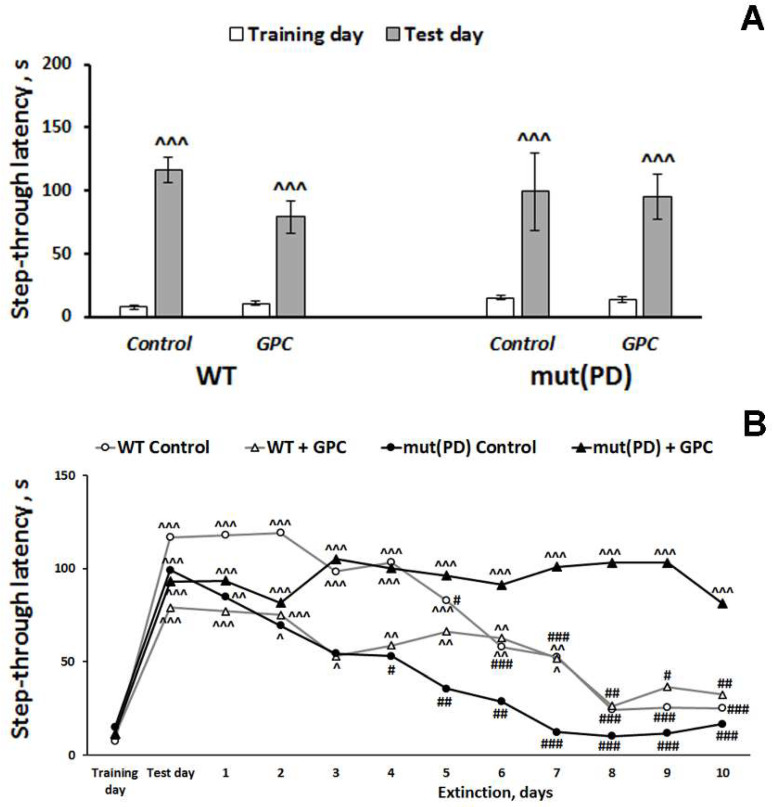
Effects of the overexpression of A53T-mutant α-synuclein and dietary supplementation with *GPC* for four months on memory retrieval (**A**) and memory extinction (**B**) in mice in the passive avoidance test. The data are expressed as the means ± SEMs (**A**) or means (**B**) of the values obtained in an independent group of animals (*n* = 5–9 per group). Statistically significant differences: ^ *p* < 0.05, ^^ *p* < 0.01, ^^^ *p* < 0.001 compared to values of the same group on the training day; ^#^
*p* < 0.05, ^##^
*p* < 0.01, ^###^
*p* < 0.001 compared to values of the same group on the test day.

**Figure 4 molecules-25-05339-f004:**
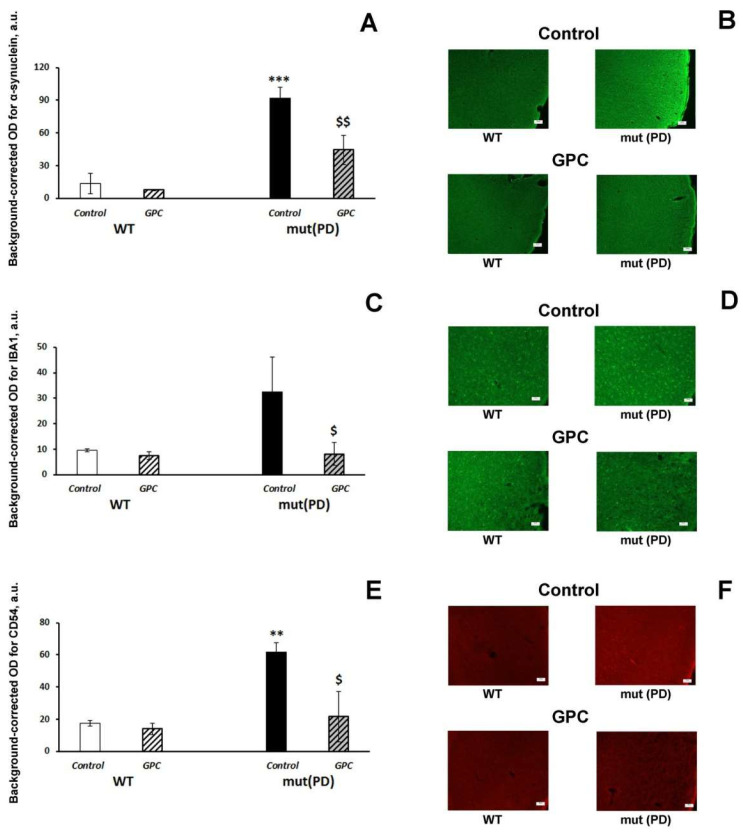
Effects of the overexpression of A53T-mutant α-synuclein and dietary supplementation with *GPC* for four months on α-synuclein accumulation (**A**,**B**) and expression of the microglial marker IBA1 (**C**,**D**) or inflammatory marker CD54 (**E**,**F**) in the frontal cortex of mice. (**A**,**C**,**E**): quantitative results. The data are expressed as the means ± SEMs of the values obtained in an independent group of animals (*n* = 3 per group). Statistically significant differences: ** *p* < 0.01, *** *p* < 0.001 vs. WT controls; ^$^
*p* < 0.05, ^$$^
*p* < 0.01 vs. mut (PD)controls. (**B**) α-synuclein immunoreactivity in the frontal cortex. (zoom: 100×; bar: 100 μm.) (**D**) IBA1 immunoreactivity in the frontal cortex. (zoom: 200×; bar; 50 μm.) (**F**) CD54 immunoreactivity in the frontal cortex. (zoom: \200×; bar: 50 μm).

**Figure 5 molecules-25-05339-f005:**
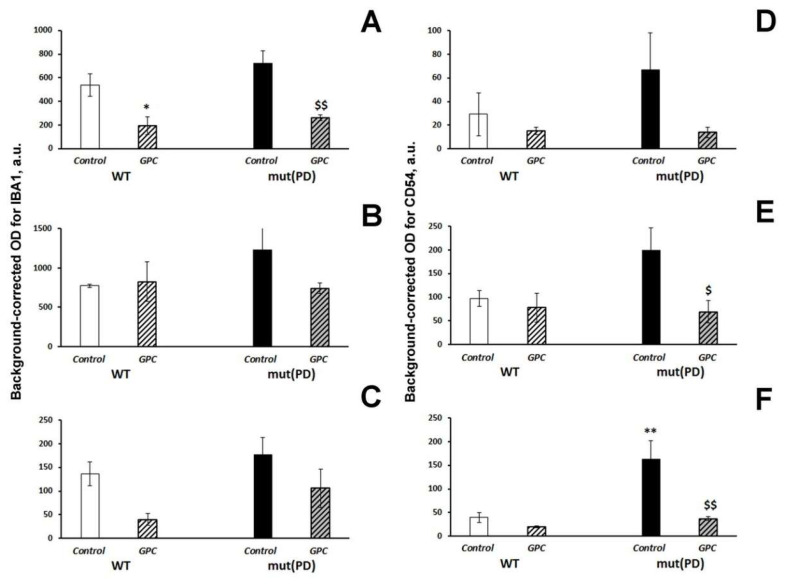
Effects of the overexpression of A53T-mutant α-synuclein and dietary supplementation with *GPC* for four months on the expression of the microglial marker IBA1 (**A**–**C**) or inflammatory marker CD54 (**D**–**F**) in the hippocampus (**A,D** in the CA1 area; **B**,**E** in the CA3 area; **C**,**F** in the dentate gyrus) in mice. The data are expressed as the means ± SEMs of the values obtained in an independent group of animals (*n* = 3 per group). Statistically significant differences: * *p* < 0.05, ** *p* < 0.01 vs. WT controls; ^$^
*p* < 0.05, ^$$^
*p* < 0.01 vs. mut(PD) controls.

**Table 1 molecules-25-05339-t001:** Effects of *GPC* supplementation and overexpression of α-synuclein (genetic Parkinson’s disease (PD) model) on the behavior of mice in the open field test.

Parameter	Group				F, *p*
	WT		Mut (PD)		
	*Control*	*GPC*	*Control*	*GPC*	
Distance travelled, cm	3773 ± 266	3704 ± 297	4964 ± 602	5236 ± 492 ^&&^	G: F(1, 28) = 10.8, *p* < 0.01
					D: F(1, 28) < 1
					D × G: F(1, 28) < 1
Rearings, n	74.7 ± 6.3	67.0 ± 8.9	76.4 ± 7.3	91.8 ± 12.3	G: F(1, 28) = 1.8, *p* > 0.05
					D: F(1, 28) < 1
					D × G: F(1, 28) = 1.4, *p* > 0.05
Time in the center, s	31.5 ± 4.8	33.2 ± 4.5	36.0 ± 9.8	30.6 ± 6.6	G: F(1, 28) < 1
					D: F(1, 28) < 1
					D × G: F(1, 28) < 1
Fecal boli, n	2.22 ± 0.74	2.89 ± 0.75	2.8 ± 1.16	1.0 ± 0.58	G: F(1, 28) < 1
					D: F(1, 28) < 1
					D × G: F(1, 28) = 2.4, *p* > 0.05

G—genotype factor, D—diet factor. Data are presented as the Mean ± S.E.M. of the values obtained in an independent group of animals (*n* = 5–9 per group). Statistically significant differences: ^&&^
*p* < 0.01 vs. a WT + GPC group.

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
