# Peer review of "Effects of Grape Polyphenols on the Life Span and Neuroinflammatory Alterations Related to Neurodegenerative Parkinson Disease-Like Disturbances in Mice"

_molecules, 2020, doi:10.3390/molecules25225339_

Round 1

Reviewer 1 Report

The authors studied the effect of polyphenols, contained in grape extracts, in animal models for Parkinson’s disease (PD). The obtained results suggest that the polyphenols of grape are able to prevent the development of PD by decreasing neuroinflammation biomarkers (CD54 and IBA1) and α-synuclein accumulation, suggesting that dietary supplementation with this natural compounds could be a promising strategy to decrease the appearance of this age-related disease.

The authors evaluated the effects of such natural compounds on a transgenic mice model for Parkinson’s disease (PD) and C57BL/6 mice were used as control in this experiment.

The economic and social burden of PD, such as for other neurodegenerative diseases, is very substantial in all countries, especially for developed countries, and reflects in high health care costs as well as loss of productivity due to morbidity and premature death. Finding prevention and therapy strategies for this class of diseases are therefore urgent. The presented work is very interesting and explores a possible strategy for PD prevention/therapy.

The manuscript is well-written and structured; however some points should be addressed to improve the manuscript. I recommend revisions before publication. Below the authors can find some suggestions and questions.

Keywords: please avoid repeating words from the title, such as grape, life span, polyphenols

Introduction:

α-synuclein is one of the main biomarkers evaluated in this study. However, in the introduction the authors do not discuss or cite previously reported works on the effect of polyphenols on the α-synuclein accumulation/aggregation in in vitro or in vivo models for PD. Several works can be found regarding this matter. Please add some information/discussion about these works, on your introduction.

Results:

Line 70 and line 90: The number of animals per group is different? Due to the death of some animals or was the experiment designed in that way? Please clarify.

Table 1: The column with statistical analysis is hard to read. Please consider dividing the column in two or more columns or add rows for better understanding.

Line 94: “The rest of the parameters studied were NOT significantly influenced ….”

Lines 97-99: This sentence is not clear. Please rewrite to clarify.

Line 126: Please write IHC name in full

Figures 4 and 5: please add info about the biomarker to the yyaxis to facilitate reading. For example, in figure 4A (Background-corrected OD for α-synuclein, a.u.).

Line 149: CA1 area instead of c,a1

Discussion: the discussion is not well supported by the obtained results. The authors should discuss in more detail the results obtained, instead just discussing previously reported works. For example, in the “open field test” the polyphenols only significantly affected the “distance travelled”. This was expected by the authors? Why? The same applies for the other conducted experiments, as for example in the “passive avoidance test”: did the authors expect these results? The same for the IHC analysis, why do the authors think that the polyphenols were able to reduce CD54 expression in the CA3 area and dentate gyrus and not in the CA1 area?

Materials and methods:

Please add information about number of animals per group in each animal experiment.

Line 270: Please write IHC name in full

Conclusions: please include some lines on future perspectives for polyphenols in PD treatment or prevention

Author Response

We would like to thank the Reviewer for the careful review of our manuscript and for his/her valuable comments and suggestions. We have thoroughly revised the manuscript considering all the comments. We believe that the revised version would be more clear and interesting for the readership of the journal.

Keywords: please avoid repeating words from the title, such as grape, life span, polyphenols

We excluded the terms used in the title and added other related to the topic. Corrected parts are marked in green.

Introduction: α-synuclein is one of the main biomarkers evaluated in this study. However, in the introduction the authors do not discuss or cite previously reported works on the effect of polyphenols on the α-synuclein accumulation/aggregation in in vitro or in vivo models for PD. Several works can be found regarding this matter. Please add some information/discussion about these works, on your introduction.

We added relevant information on α-synuclein to Introduction and Discussion. Corrected parts are marked in green.

Results:

Line 70 and line 90: The number of animals per group is different? Due to the death of some animals or was the experiment designed in that way? Please clarify.

Due to the death of some animals and some technical reasons. For instance, mice of the C57Bl/6J strain are characterized by the high level of intermale aggression. When C57Bl/6J males are housed in groups, they often fight for social hierarchy and injure each other, especially in some groups. To reduce the variability in the groups due to this factor, we had to take more mice of control group in the experiment with the life expectancy.

Table 1: The column with statistical analysis is hard to read. Please consider dividing the column in two or more columns or add rows for better understanding.

We corrected the Table according to the Reviewer’s suggestion.

Line 94: “The rest of the parameters studied were NOT significantly influenced ….”

We corrected this sentence.

Lines 97-99: This sentence is not clear. Please rewrite to clarify.

Done.

Line 126: Please write IHC name in full

Done.

Figures 4 and 5: please add info about the biomarker to the yyaxis to facilitate reading. For example, in figure 4A (Background-corrected OD for α-synuclein, a.u.).

Done.

Line 149: CA1 area instead of c,a1

Done.

Discussion: the discussion is not well supported by the obtained results. The authors should discuss in more detail the results obtained, instead just discussing previously reported works. For example, in the “open field test” the polyphenols only significantly affected the “distance travelled”. This was expected by the authors? Why? The same applies for the other conducted experiments, as for example in the “passive avoidance test”: did the authors expect these results? The same for the IHC analysis, why do the authors think that the polyphenols were able to reduce CD54 expression in the CA3 area and dentate gyrus and not in the CA1 area?

We thoroughly revised the Discussion with a focus on the obtained results.

Materials and methods:

Please add information about number of animals per group in each animal experiment.

Done.

Line 270: Please write IHC name in full

Done.

Conclusions: please include some lines on future perspectives for polyphenols in PD treatment or prevention

Done.

We would like to thank the Reviewer for the careful review of our manuscript and for his/her valuable comments and suggestions. We have thoroughly revised the manuscript considering all the comments. We believe that the revised version would be more clear and interesting for the readership of the journal.

Keywords: please avoid repeating words from the title, such as grape, life span, polyphenols

We excluded the terms used in the title and added other related to the topic. Corrected parts are marked in green.

Introduction: α-synuclein is one of the main biomarkers evaluated in this study. However, in the introduction the authors do not discuss or cite previously reported works on the effect of polyphenols on the α-synuclein accumulation/aggregation in in vitro or in vivo models for PD. Several works can be found regarding this matter. Please add some information/discussion about these works, on your introduction.

We added relevant information on α-synuclein to Introduction and Discussion. Corrected parts are marked in green.

Results:

Line 70 and line 90: The number of animals per group is different? Due to the death of some animals or was the experiment designed in that way? Please clarify.

Due to the death of some animals and some technical reasons. For instance, mice of the C57Bl/6J strain are characterized by the high level of intermale aggression. When C57Bl/6J males are housed in groups, they often fight for social hierarchy and injure each other, especially in some groups. To reduce the variability in the groups due to this factor, we had to take more mice of control group in the experiment with the life expectancy.

Table 1: The column with statistical analysis is hard to read. Please consider dividing the column in two or more columns or add rows for better understanding.

We corrected the Table according to the Reviewer’s suggestion.

Line 94: “The rest of the parameters studied were NOT significantly influenced ….”

We corrected this sentence.

Lines 97-99: This sentence is not clear. Please rewrite to clarify.

Done.

Line 126: Please write IHC name in full

Done.

Figures 4 and 5: please add info about the biomarker to the yyaxis to facilitate reading. For example, in figure 4A (Background-corrected OD for α-synuclein, a.u.).

Done.

Line 149: CA1 area instead of c,a1

Done.

Discussion: the discussion is not well supported by the obtained results. The authors should discuss in more detail the results obtained, instead just discussing previously reported works. For example, in the “open field test” the polyphenols only significantly affected the “distance travelled”. This was expected by the authors? Why? The same applies for the other conducted experiments, as for example in the “passive avoidance test”: did the authors expect these results? The same for the IHC analysis, why do the authors think that the polyphenols were able to reduce CD54 expression in the CA3 area and dentate gyrus and not in the CA1 area?

We thoroughly revised the Discussion with a focus on the obtained results.

Materials and methods:

Please add information about number of animals per group in each animal experiment.

Done.

Line 270: Please write IHC name in full

Done.

Conclusions: please include some lines on future perspectives for polyphenols in PD treatment or prevention

Done.

Reviewer 2 Report

This article is really interesting regarding to its main objetives as well as results and conclusions. This way, grape polyphenols would be reallu valuable in order to treat neurodegenerative damage like those associated to Parkinson Disease. However, authors should improve and provide some data to a better support their conclusions. I recommend a major revision of the present work, particularly to the following items:

  1. Why does the treatment with grape polyphenols have not the same time duration (4 months) in both mice models?
  2. ¿Have a shorter treatment (4 months) any effect on the survival of C57BL/6 mice?
  3. Given these premises, the effect of genotype should not be considered in the statistical analysis. The results on each animal model should be independently considered.
  4. Statistical differences of figure 3 should be represented by using asterisks.
  5. Immunofluorescence (IF) images of alpha-synuclein, iba1 and CD54 must be shown in the article, always showing the best images of the results obtained, besides theis quantification. They should show not areas of interest of the results but specific marcages of the each IF.
  6. There is some mistakes in legend of figure 5. They are noted some zones of the hippocampus, but there are not showing any image.

Author Response

We would like to thank the Reviewer for careful and detailed review of the manuscript and for his/her valuable comments and suggestions. We have thoroughly revised the manuscript considering all the comments. We believe that the revised version would be more clear and interesting for the readership of the journal.

Why does the treatment with grape polyphenols have not the same time duration (4 months) in both mice models? Have a shorter treatment (4 months) any effect on the survival of C57BL/6 mice?

There were two experiments. In the first one, we evaluated the effects of the treatment with grape polyphenols on the life span in mice of C57Bl/6 strain. Since the average life expectancy in mice upon normal conditions is much longer than four months, we have not limited the diet supplementation with grape polyphenols by certain period but chosen the life-long treatment design. We have not observed any effect on the survival of WT mice after four months of treatment in the second experiment. All WT mice were alive. In the second experiment, we studied the effects of grape polyphenols on the early development of PD-like pathology. The duration of treatment in the experiment was chosen as the transgenic PD mice used in the study manifest certain behavioral and neurochemical PD-like alterations since the age of 5-7 months.

Given these premises, the effect of genotype should not be considered in the statistical analysis. The results on each animal model should be independently considered.

The experiments were analyzed independently. In the experiment with the effects on life span, there was only one model (mice of C57Bl/6). In the experiment with the PD model, we focused on another model, mice of B6.Cg-Tg(Prnp-SNCA*A53T)23Mkle/J) strain, a transgenic PD model. WT mice were used as an adequate genetic control to the mice of PD model. They have the same genotype except for the transgene encoding human mutant alpha-synuclein. That allowed us monitoring the effects of transgene and evaluating possible interaction between the factors (transgene and diet).

Statistical differences of figure 3 should be represented by using asterisks.

We used asterisks to designate the differences between WT and mut(PD) in the experiment with the PD model. We suppose that it would be more clear for readers if the same signs for the same type of comparisons are used in all figures.

Immunofluorescence (IF) images of alpha-synuclein, iba1 and CD54 must be shown in the article, always showing the best images of the results obtained, besides theis quantification. They should show not areas of interest of the results but specific marcages of the each IF.

We revised the figures presenting the results of immunohistochemical analysis. We added IBA1 and CD54 images to Figure 4. We also add a Figure S1 with IBA1 and CD54 images corresponding to the diagrams in the Figure 5.

There is some mistakes in legend of figure 5. They are noted some zones of the hippocampus, but there are not showing any image.

We corrected the figure legend.

We would like to thank the Reviewer for careful and detailed review of the manuscript and for his/her valuable comments and suggestions. We have thoroughly revised the manuscript considering all the comments. We believe that the revised version would be more clear and interesting for the readership of the journal.

Why does the treatment with grape polyphenols have not the same time duration (4 months) in both mice models? Have a shorter treatment (4 months) any effect on the survival of C57BL/6 mice?

There were two experiments. In the first one, we evaluated the effects of the treatment with grape polyphenols on the life span in mice of C57Bl/6 strain. Since the average life expectancy in mice upon normal conditions is much longer than four months, we have not limited the diet supplementation with grape polyphenols by certain period but chosen the life-long treatment design. We have not observed any effect on the survival of WT mice after four months of treatment in the second experiment. All WT mice were alive. In the second experiment, we studied the effects of grape polyphenols on the early development of PD-like pathology. The duration of treatment in the experiment was chosen as the transgenic PD mice used in the study manifest certain behavioral and neurochemical PD-like alterations since the age of 5-7 months.

Given these premises, the effect of genotype should not be considered in the statistical analysis. The results on each animal model should be independently considered.

The experiments were analyzed independently. In the experiment with the effects on life span, there was only one model (mice of C57Bl/6). In the experiment with the PD model, we focused on another model, mice of B6.Cg-Tg(Prnp-SNCA*A53T)23Mkle/J) strain, a transgenic PD model. WT mice were used as an adequate genetic control to the mice of PD model. They have the same genotype except for the transgene encoding human mutant alpha-synuclein. That allowed us monitoring the effects of transgene and evaluating possible interaction between the factors (transgene and diet).

Statistical differences of figure 3 should be represented by using asterisks.

We used asterisks to designate the differences between WT and mut(PD) in the experiment with the PD model. We suppose that it would be more clear for readers if the same signs for the same type of comparisons are used in all figures.

Immunofluorescence (IF) images of alpha-synuclein, iba1 and CD54 must be shown in the article, always showing the best images of the results obtained, besides theis quantification. They should show not areas of interest of the results but specific marcages of the each IF.

We revised the figures presenting the results of immunohistochemical analysis. We added IBA1 and CD54 images to Figure 4. We also add a Figure S1 with IBA1 and CD54 images corresponding to the diagrams in the Figure 5.

There is some mistakes in legend of figure 5. They are noted some zones of the hippocampus, but there are not showing any image.

We corrected the figure legend.

We would like to thank the Reviewer for careful and detailed review of the manuscript and for his/her valuable comments and suggestions. We have thoroughly revised the manuscript considering all the comments. We believe that the revised version would be more clear and interesting for the readership of the journal.

Why does the treatment with grape polyphenols have not the same time duration (4 months) in both mice models? Have a shorter treatment (4 months) any effect on the survival of C57BL/6 mice?

There were two experiments. In the first one, we evaluated the effects of the treatment with grape polyphenols on the life span in mice of C57Bl/6 strain. Since the average life expectancy in mice upon normal conditions is much longer than four months, we have not limited the diet supplementation with grape polyphenols by certain period but chosen the life-long treatment design. We have not observed any effect on the survival of WT mice after four months of treatment in the second experiment. All WT mice were alive. In the second experiment, we studied the effects of grape polyphenols on the early development of PD-like pathology. The duration of treatment in the experiment was chosen as the transgenic PD mice used in the study manifest certain behavioral and neurochemical PD-like alterations since the age of 5-7 months.

Given these premises, the effect of genotype should not be considered in the statistical analysis. The results on each animal model should be independently considered.

The experiments were analyzed independently. In the experiment with the effects on life span, there was only one model (mice of C57Bl/6). In the experiment with the PD model, we focused on another model, mice of B6.Cg-Tg(Prnp-SNCA*A53T)23Mkle/J) strain, a transgenic PD model. WT mice were used as an adequate genetic control to the mice of PD model. They have the same genotype except for the transgene encoding human mutant alpha-synuclein. That allowed us monitoring the effects of transgene and evaluating possible interaction between the factors (transgene and diet).

Statistical differences of figure 3 should be represented by using asterisks.

We used asterisks to designate the differences between WT and mut(PD) in the experiment with the PD model. We suppose that it would be more clear for readers if the same signs for the same type of comparisons are used in all figures.

Immunofluorescence (IF) images of alpha-synuclein, iba1 and CD54 must be shown in the article, always showing the best images of the results obtained, besides theis quantification. They should show not areas of interest of the results but specific marcages of the each IF.

We revised the figures presenting the results of immunohistochemical analysis. We added IBA1 and CD54 images to Figure 4. We also add a Figure S1 with IBA1 and CD54 images corresponding to the diagrams in the Figure 5.

There is some mistakes in legend of figure 5. They are noted some zones of the hippocampus, but there are not showing any image.

We corrected the figure legend.

We would like to thank the Reviewer for careful and detailed review of the manuscript and for his/her valuable comments and suggestions. We have thoroughly revised the manuscript considering all the comments. We believe that the revised version would be more clear and interesting for the readership of the journal.

Why does the treatment with grape polyphenols have not the same time duration (4 months) in both mice models? Have a shorter treatment (4 months) any effect on the survival of C57BL/6 mice?

There were two experiments. In the first one, we evaluated the effects of the treatment with grape polyphenols on the life span in mice of C57Bl/6 strain. Since the average life expectancy in mice upon normal conditions is much longer than four months, we have not limited the diet supplementation with grape polyphenols by certain period but chosen the life-long treatment design. We have not observed any effect on the survival of WT mice after four months of treatment in the second experiment. All WT mice were alive. In the second experiment, we studied the effects of grape polyphenols on the early development of PD-like pathology. The duration of treatment in the experiment was chosen as the transgenic PD mice used in the study manifest certain behavioral and neurochemical PD-like alterations since the age of 5-7 months.

Given these premises, the effect of genotype should not be considered in the statistical analysis. The results on each animal model should be independently considered.

The experiments were analyzed independently. In the experiment with the effects on life span, there was only one model (mice of C57Bl/6). In the experiment with the PD model, we focused on another model, mice of B6.Cg-Tg(Prnp-SNCA*A53T)23Mkle/J) strain, a transgenic PD model. WT mice were used as an adequate genetic control to the mice of PD model. They have the same genotype except for the transgene encoding human mutant alpha-synuclein. That allowed us monitoring the effects of transgene and evaluating possible interaction between the factors (transgene and diet).

Statistical differences of figure 3 should be represented by using asterisks.

We used asterisks to designate the differences between WT and mut(PD) in the experiment with the PD model. We suppose that it would be more clear for readers if the same signs for the same type of comparisons are used in all figures.

Immunofluorescence (IF) images of alpha-synuclein, iba1 and CD54 must be shown in the article, always showing the best images of the results obtained, besides theis quantification. They should show not areas of interest of the results but specific marcages of the each IF.

We revised the figures presenting the results of immunohistochemical analysis. We added IBA1 and CD54 images to Figure 4. We also add a Figure S1 with IBA1 and CD54 images corresponding to the diagrams in the Figure 5.

There is some mistakes in legend of figure 5. They are noted some zones of the hippocampus, but there are not showing any image.

We corrected the figure legend.

We would like to thank the Reviewer for careful and detailed review of the manuscript and for his/her valuable comments and suggestions. We have thoroughly revised the manuscript considering all the comments. We believe that the revised version would be more clear and interesting for the readership of the journal.

Why does the treatment with grape polyphenols have not the same time duration (4 months) in both mice models? Have a shorter treatment (4 months) any effect on the survival of C57BL/6 mice?

There were two experiments. In the first one, we evaluated the effects of the treatment with grape polyphenols on the life span in mice of C57Bl/6 strain. Since the average life expectancy in mice upon normal conditions is much longer than four months, we have not limited the diet supplementation with grape polyphenols by certain period but chosen the life-long treatment design. We have not observed any effect on the survival of WT mice after four months of treatment in the second experiment. All WT mice were alive. In the second experiment, we studied the effects of grape polyphenols on the early development of PD-like pathology. The duration of treatment in the experiment was chosen as the transgenic PD mice used in the study manifest certain behavioral and neurochemical PD-like alterations since the age of 5-7 months.

Given these premises, the effect of genotype should not be considered in the statistical analysis. The results on each animal model should be independently considered.

The experiments were analyzed independently. In the experiment with the effects on life span, there was only one model (mice of C57Bl/6). In the experiment with the PD model, we focused on another model, mice of B6.Cg-Tg(Prnp-SNCA*A53T)23Mkle/J) strain, a transgenic PD model. WT mice were used as an adequate genetic control to the mice of PD model. They have the same genotype except for the transgene encoding human mutant alpha-synuclein. That allowed us monitoring the effects of transgene and evaluating possible interaction between the factors (transgene and diet).

Statistical differences of figure 3 should be represented by using asterisks.

We used asterisks to designate the differences between WT and mut(PD) in the experiment with the PD model. We suppose that it would be more clear for readers if the same signs for the same type of comparisons are used in all figures.

Immunofluorescence (IF) images of alpha-synuclein, iba1 and CD54 must be shown in the article, always showing the best images of the results obtained, besides theis quantification. They should show not areas of interest of the results but specific marcages of the each IF.

We revised the figures presenting the results of immunohistochemical analysis. We added IBA1 and CD54 images to Figure 4. We also add a Figure S1 with IBA1 and CD54 images corresponding to the diagrams in the Figure 5.

There is some mistakes in legend of figure 5. They are noted some zones of the hippocampus, but there are not showing any image.

We corrected the figure legend.

We would like to thank the Reviewer for careful and detailed review of the manuscript and for his/her valuable comments and suggestions. We have thoroughly revised the manuscript considering all the comments. We believe that the revised version would be more clear and interesting for the readership of the journal.

Why does the treatment with grape polyphenols have not the same time duration (4 months) in both mice models? Have a shorter treatment (4 months) any effect on the survival of C57BL/6 mice?

There were two experiments. In the first one, we evaluated the effects of the treatment with grape polyphenols on the life span in mice of C57Bl/6 strain. Since the average life expectancy in mice upon normal conditions is much longer than four months, we have not limited the diet supplementation with grape polyphenols by certain period but chosen the life-long treatment design. We have not observed any effect on the survival of WT mice after four months of treatment in the second experiment. All WT mice were alive. In the second experiment, we studied the effects of grape polyphenols on the early development of PD-like pathology. The duration of treatment in the experiment was chosen as the transgenic PD mice used in the study manifest certain behavioral and neurochemical PD-like alterations since the age of 5-7 months.

Given these premises, the effect of genotype should not be considered in the statistical analysis. The results on each animal model should be independently considered.

The experiments were analyzed independently. In the experiment with the effects on life span, there was only one model (mice of C57Bl/6). In the experiment with the PD model, we focused on another model, mice of B6.Cg-Tg(Prnp-SNCA*A53T)23Mkle/J) strain, a transgenic PD model. WT mice were used as an adequate genetic control to the mice of PD model. They have the same genotype except for the transgene encoding human mutant alpha-synuclein. That allowed us monitoring the effects of transgene and evaluating possible interaction between the factors (transgene and diet).

Statistical differences of figure 3 should be represented by using asterisks.

We used asterisks to designate the differences between WT and mut(PD) in the experiment with the PD model. We suppose that it would be more clear for readers if the same signs for the same type of comparisons are used in all figures.

Immunofluorescence (IF) images of alpha-synuclein, iba1 and CD54 must be shown in the article, always showing the best images of the results obtained, besides theis quantification. They should show not areas of interest of the results but specific marcages of the each IF.

We revised the figures presenting the results of immunohistochemical analysis. We added IBA1 and CD54 images to Figure 4. We also add a Figure S1 with IBA1 and CD54 images corresponding to the diagrams in the Figure 5.

There is some mistakes in legend of figure 5. They are noted some zones of the hippocampus, but there are not showing any image.

We corrected the figure legend.

Reviewer 3 Report

Tikhonova et al. have assessed the effects of supplementation of the diet with grape polyphenols in a model of Parkinson’s disease. They report that this supplementation increases the lifespan of the control mice, enhances memory, and reduces the expression of α-synuclein and inflammatory markers in the transgenic model mice.

This manuscript is written relatively poorly. I do have some comments and questions that I think will help the authors improve the text.

Major comments:

  1. In the methods (and throughout the manuscript), the authors must rephrase the description of the dosage of GPC given to mice. If GPC were available to the mice ad libitum, then the dose cannot be precisely described as 1.5 mL/kg. In fact, 1.5 mL/kg does not clearly describe the dosage to this reviewer, and it is confusing and mostly meaningless. As I understand it, the mice had access to supplemented water ad libitum; thus, the mice of any weight could drink as much or as little water as they wanted per day. Thus, the reported dosage does not mean anything, especially when mentioning per bodyweight of the mice; this aspect could not have been controlled by ad libitum access to water. The dose 1.5 mL/kg would have meant clearly if this were an injection, controlled by the experimenter. A meaningful way would be to describe how much GPC (volume) was added to a volume of the drinking water (volume/volume). If drinking were ad libitum, there is no way that the per-weight aspect could have been controlled. See lines 23 and 240.
  2. I would like the authors to consider the possibility that many components of the GPC mixture may act as pan-assay-interference (PAINS) compounds (see PMID: 25254460, PMID: 32847100). Evidence suggests that EGCG, resveratrol, and quercetin are pan-assay interfering compounds (PAIN). I would encourage the authors to be critical of the literature when presenting their data and discuss the potential limitations. See, ‘Kanlaya R, Thongboonkerd V. Molecular Mechanisms of Epigallocatechin-3-Gallate for Prevention of Chronic Kidney Disease and Renal Fibrosis: Preclinical Evidence. Curr Dev Nutr. 2019 Aug 29;3(9):nzz101. doi: 10.1093/cdn/nzz101. PMID: 31555758; PMCID: PMC6752729.’ See https://blogs.sciencemag.org/pipeline/archives/2015/11/11/screen-carefully; and Helgi I. Ingólfsson, Pratima Thakur, Karl F. Herold, E. Ashley Hobart, Nicole B. Ramsey, Xavier Periole, Djurre H. de Jong, Martijn Zwama, Duygu Yilmaz, Katherine Hall, Thorsten Maretzky, Hugh C. Hemmings, Carl Blobel, Siewert J. Marrink, Armağan Koçer, Jon T. Sack, and Olaf S. Andersen ACS Chemical Biology 2014 9 (8), 1788-1798 DOI: 10.1021/cb500086e.
    What experiments and data presented in this manuscript may provide evidence against this possibility of PAINS and the fact that many different metabolites of the polyphenols may function as diverse pathway modulators instead of the polyphenols (see my comment regarding the blood–brain barrier). I would like the authors to discuss this and highlight the evidence I pointed out, so they present an unbiased view of the literature.

Rest of the manuscript:

  1. The range of numbers must be separated by an en-dash. Compounds are only hyphenated when acting as adjectival phrases. For example, in line 24, change ‘since the age of 1.5-2-month-old’ to ‘since the age of 1.5–2 months’. The word ‘old’ is not necessary. The same should apply to line 25, lines 73 and 72 (delete ‘old’), line 243, and line 245.
  2. In line 31, change ‘at’ to ‘during’.
  3. In line 47, add ‘of’ after ‘range’ (missing preposition).
  4. Rephrase the first clause in line 49. It does not make any sense. In the same line, change ‘grapes seeds’ to ‘grape seed’. Please do not use a plural form as an adjective. Revise this throughout the manuscript. See line 54, ‘red grape flavonoids’ instead of ‘red grapes’.
  5. In line 56 (and throughout the manuscript), please revise the reporting of the previous literature and provide sufficient and relevant details. In this line, the referral to the blood–brain barrier (note the use of en-dash between blood and brain instead of hyphen), applies to a study reported in rats (reference 17). Report this specifically, and do not generalise. As it is written, the statement is misleading considering the previous literature cited. If there is evidence in humans, please refer to those in this context. The reference 17 is not a piece of direct evidence. The reference 16 cited within reference 17 uses an in situ model of the blood–brain barrier in a co-culture system. Thus, it seems, there is no direct evidence that intact polyphenols can cross the barrier in an animal or in humans (?). Besides, it has been discussed that the small-molecule metabolites of polyphenols likely are the compounds that can cross the barrier. Considering these facts, please revise the reporting for factual accuracy and provide indisputable evidence when reporting on the fact that polyphenols can cross the blood–brain barrier in the study model or in humans. See PMID: 22012276 and PMID: 28904352 on PubMed.
  6. Line 51: In academic writing, ‘etc.’ is best avoided as it does not add any value or information; it is a vacuum. It is best to be informative and specific in the text, instead of using ‘etc.’ and avoid using it.
  7. In multi-panel figures, increase the font to make the numbers readable. For example, see Figure 1A, the numbers for the x-axis and the y-axis.
  8. Use the decimal point (for international English) instead of a comma (European style) to separate the whole number from the fractional part of the number (Figure 1A, Table S1).
  9. Table S1 and Supplementary File 1 are the same. There is a mix-up in the text while referring to these. Please correct this and remove this duplication.
  10. Is there a difference between Enoant and GPC? Seemingly, Enoant and GPC have been used interchangeably in the text (or figures) and can be confusing. For consistency, use one of them throughout the manuscript.
  11. The data are best to be presented as mean ± SD instead of mean ± SEM. Revise throughout the manuscript.
  12. In line 85, change ‘estimate’ to ‘examine’ or ‘assess’. Revise the word choice throughout the manuscript.
  13. Rephrase the sentence in line 94.
  14. In line 149, correct CA1.
  15. In 184, change ‘accompanying’ to ‘accompanied’.
  16. In 187, change ‘effect’ to ‘affect’.
  17. In line 217, the authors mention, ‘we can suggest multiple cell mechanisms of the GP-diet effect’, however, they do not mention these mechanisms, nor they provide direct evidence on these mechanisms. Please revise this speculation.
  18. In line 259, the parenthetical (1000 1x) and in line 252 (28 1x) do not mean anything to this reviewer. Could you clarify these in the text, please?
  19. Rephrase the sentence in line 271. The mice were not anesthetized by CO2; as I understand, they would have been culled by CO2. See PMID: 16996534.
  20. The literature-dominating apostrophe form of ‘Parkinson’s Disease’ is logically wrong. The disease did not belong to Parkinson; he did not suffer from it; he merely described it. The preferred designation would be ‘Parkinson Disease’. See the Chicago Manual of Style and The Australian Manual of Scientific Style. Both these style manuals suggest using ‘Parkinson Disease’ instead of ‘Parkinson’s disease’.
  21. English syntax, word choice, and punctuation must be improved in the text. Overall, the English in the manuscript will benefit from proper editing and proofreading.

Author Response

In the methods (and throughout the manuscript), the authors must rephrase the description of the dosage of GPC given to mice. If GPC were available to the mice ad libitum, then the dose cannot be precisely described as 1.5 mL/kg. In fact, 1.5 mL/kg does not clearly describe the dosage to this reviewer, and it is confusing and mostly meaningless. As I understand it, the mice had access to supplemented water ad libitum; thus, the mice of any weight could drink as much or as little water as they wanted per day. Thus, the reported dosage does not mean anything, especially when mentioning per bodyweight of the mice; this aspect could not have been controlled by ad libitum access to water. The dose 1.5 mL/kg would have meant clearly if this were an injection, controlled by the experimenter. A meaningful way would be to describe how much GPC (volume) was added to a volume of the drinking water (volume/volume). If drinking were ad libitum, there is no way that the per-weight aspect could have been controlled. See lines 23 and 240.

The daily averaged dosage of Enoant was 1.5 mL/kg. Mice were weekly weighed and the liquid consumption per cage was measured to correct the dosage. Hence, the amount of GPC dissolved in the pure water varied during the experiment (0.5-0.8%) depending on the mouse weight and water consumption. We added the detailed description to the Methods. Corrected parts are marked in green.

I would like the authors to consider the possibility that many components of the GPC mixture may act as pan-assay-interference (PAINS) compounds (see PMID: 25254460, PMID: 32847100). Evidence suggests that EGCG, resveratrol, and quercetin are pan-assay interfering compounds (PAIN). I would encourage the authors to be critical of the literature when presenting their data and discuss the potential limitations. See, ‘Kanlaya R, Thongboonkerd V. Molecular Mechanisms of Epigallocatechin-3-Gallate for Prevention of Chronic Kidney Disease and Renal Fibrosis: Preclinical Evidence. Curr Dev Nutr. 2019 Aug 29;3(9):nzz101. doi: 10.1093/cdn/nzz101. PMID: 31555758; PMCID:PMC6752729.’ See https://blogs.sciencemag.org/pipeline/archives/2015/11/11/screen-carefully; and Helgi I. Ingólfsson, Pratima Thakur, Karl F. Herold, E. Ashley Hobart, Nicole B. Ramsey, Xavier Periole, Djurre H. de Jong, Martijn Zwama, Duygu Yilmaz, Katherine Hall, Thorsten Maretzky, Hugh C. Hemmings, Carl Blobel, Siewert J. Marrink, Armağan Koçer, Jon T. Sack, and Olaf S. Andersen ACS Chemical Biology 2014 9 (8), 1788-1798 DOI: 10.1021/cb500086e.

What experiments and data presented in this manuscript may provide evidence against this possibility of PAINS and the fact that many different metabolites of the polyphenols may function as diverse pathway modulators instead of the polyphenols (see my comment regarding the blood–brain barrier). I would like the authors to discuss this and highlight the evidence I pointed out, so they present an unbiased view of the literature.

We revised the Discussion and touched on the issues of polyphenol bioavailability, blood-brain barrier crossing, and PAINS properties.

Rest of the manuscript:

The range of numbers must be separated by an en-dash. Compounds are only hyphenated when acting as adjectival phrases. For example, in line 24, change ‘since the age of 1.5-2-month-old’ to ‘since the age of 1.5–2 months’. The word ‘old’ is not necessary. The same should apply to line 25, lines 73 and 72 (delete ‘old’), line 243, and line 245.

Corrected.

In line 31, change ‘at’ to ‘during’.

Corrected.

In line 47, add ‘of’ after ‘range’ (missing preposition).

Done.

Rephrase the first clause in line 49. It does not make any sense. In the same line, change ‘grapes seeds’ to ‘grape seed’. Please do not use a plural form as an adjective. Revise this throughout the manuscript. See line 54, ‘red grape flavonoids’ instead of ‘red grapes’.

Corrected.

In line 56 (and throughout the manuscript), please revise the reporting of the previous literature and provide sufficient and relevant details. In this line, the referral to the blood–brain barrier (note the use of en-dash between blood and brain instead of hyphen), applies to a study reported in rats (reference 17). Report this specifically, and do not generalise. As it is written, the statement is misleading considering the previous literature cited. If there is evidence in humans, please refer to those in this context. The reference 17 is not a piece of direct evidence. The reference 16 cited within reference 17 uses an in situ model of the blood–brain barrier in a co-culture system. Thus, it seems, there is no direct evidence that intact polyphenols can cross the barrier in an animal or in humans (?). Besides, it has been discussed that the small-molecule metabolites of polyphenols likely are the compounds that can cross the barrier. Considering these facts, please revise the reporting for factual accuracy and provide indisputable evidence when reporting on the fact that polyphenols can cross the blood–brain barrier in the study model or in humans. See PMID: 22012276 and PMID: 28904352 on PubMed.

We revised that passage and references cited.

Line 51: In academic writing, ‘etc.’ is best avoided as it does not add any value or information; it is a vacuum. It is best to be informative and specific in the text, instead of using ‘etc.’ and avoid using it.

Done.

In multi-panel figures, increase the font to make the numbers readable. For example, see Figure 1A, the numbers for the x-axis and the y-axis. Use the decimal point (for international English) instead of a comma (European style) to separate the whole number from the fractional part of the number (Figure 1A, Table S1).

Corrected.

Table S1 and Supplementary File 1 are the same. There is a mix-up in the text while referring to these. Please correct this and remove this duplication.

Corrected.

Is there a difference between Enoant and GPC? Seemingly, Enoant and GPC have been used interchangeably in the text (or figures) and can be confusing. For consistency, use one of them throughout the manuscript.

Corrected.

The data are best to be presented as mean ± SD instead of mean ± SEM. Revise throughout the manuscript.

We cannot agree with the Reviewer that the data presentation as mean ± SD is better than mean ± SEM. Both ways are adequate, acceptable, and commonly used in biomedical research papers.

In line 85, change ‘estimate’ to ‘examine’ or ‘assess’. Revise the word choice throughout the manuscript.

Done.

Rephrase the sentence in line 94.

Done.

In line 149, correct CA1.

Done.

In 184, change ‘accompanying’ to ‘accompanied’.

Done.

In line 217, the authors mention, ‘we can suggest multiple cell mechanisms of the GP-diet effect’, however, they do not mention these mechanisms, nor they provide direct evidence on these mechanisms. Please revise this speculation.

The passage was revised.

In line 259, the parenthetical (1000 1x) and in line 252 (28 1x) do not mean anything to this reviewer. Could you clarify these in the text, please?

Lx is a symbol for Lux (SI derived unit of illuminance).

Rephrase the sentence in line 271. The mice were not anesthetized by CO2; as I understand, they would have been culled by CO2. See PMID: 16996534.

Done.

The literature-dominating apostrophe form of ‘Parkinson’s Disease’ is logically wrong. The disease did not belong to Parkinson; he did not suffer from it; he merely described it. The preferred designation would be ‘Parkinson Disease’. See the Chicago Manual of Style and The Australian Manual of Scientific Style. Both these style manuals suggest using ‘Parkinson Disease’ instead of ‘Parkinson’s disease’.

We corrected the term according to the mentioned style manuals.

English syntax, word choice, and punctuation must be improved in the text. Overall, the English in the manuscript will benefit from proper editing and proofreading.

We proofread the manuscript and did our best to improve the text

Round 2

Reviewer 2 Report

Thanks to the authors for these valuable corrections in order to improve the quality presentation of their work. 

Author Response

Thanks for evaluating our work

Reviewer 3 Report

I thank the authors for thoroughly revising their manuscript. I have one issue I would like to raise with the authors about the dosage. The authors mention in their response to the reviewers and in the text that the dosage was corrected; see quoted text: 'The daily averaged dosage of Enoant was 1.5 mL/kg. Mice were weekly weighed and the liquid consumption per cage was measured to correct the dosage. Hence, the amount of GPC dissolved in the pure water varied during the experiment (0.5-0.8%) depending on the mouse weight and water consumption. We added the detailed description to the Methods. Corrected parts are marked in green.' The word 'corrected' for dosage does not make sense. How can the amount of the compound be corrected when the mouse has already been dosed with the compound? How could 'the daily average dose' be determined or corrected if the mice were weighed and liquid consumption per cage was measured 'every week'? Does this mean that the mice were dosed for a week before it was determined how much they took of the compound and then that amount was determined to assess the daily average? I suggest the authors describe this better and change the word 'corrected' to 'adjusted'. I think they mean applying adjustments rather then 'corrections'.

I suggest the authors should thoroughly proofread the newly added text. There are some loose English-usage issues in the newly added text while the revised text reads better than what was presented in the previous version. 

Author Response

We would like to thank the Reviewer for the review of the revised manuscript and for his/her valuable comments. We corrected again the description of GPC dosage. We hope that the revised version would be more clear for the Reviewer and readership of the journal. The mice of GPC groups were given the standard chow and aqueous solution of the concentrate of grape polyphenols Enoant (concentrate produced by company RESSFOOD, Yalta, Russia, http://www.enoant.info) ad libitumEnoant was diluted with the pure water to a concentration of 0.5-0.8% taking into account a mouse body weight and liquid consumption to provide the daily averaged dosage of 1.5 mL/kg. Fresh solution was prepared every other day. To adjust the dosage, liquid consumption per cage was measured daily and mice were weighed weekly. We agree with the Reviewer that the word 'adjust' would describe the procedure better. We changed the word 'correct' for 'adjust'. We mean that we calculated the dilution when preparing fresh solution every other day to achieve the daily averaged dosage of 1.5 mL/kg. We took into account mouse body weight measured weekly and daily liquid consumption per cage. The parameters changed slowly, hence the dose consumed actually did not vary much from the expected averaged dose.